# The Impact of Sociodemographic Characteristics on Coping Strategies Used by Nurses Working at COVID and Non-COVID Hospital Departments during COVID-19 Pandemic: A Cross-Sectional Study

**DOI:** 10.3390/healthcare10061144

**Published:** 2022-06-20

**Authors:** Matea Dolić, Vesna Antičević, Krešimir Dolić, Zenon Pogorelić

**Affiliations:** 1Department of Health Studies, University of Split, 21000 Split, Croatia; vesna.anticevic@ozs.unist.hr (V.A.); kdolic79@gmail.com (K.D.); 2Department of Diagnostic and Interventional Radiology, University Hospital of Split, 21000 Split, Croatia; 3School of Medicine, University of Split, 21000 Split, Croatia; zpogorelic@gmail.com; 4Department of Pediatric Surgery, University Hospital of Split, 21000 Split, Croatia

**Keywords:** COVID-19 pandemic, nurse, coping strategies, sociodemographic factors

## Abstract

Background: The aim of our study was to compare coping strategies applied by nurses working during the COVID-19 pandemic at COVID-19 (CoV) and non-COVID-19 (non-CoV) hospital departments with regards to their sociodemographic characteristics in order that the system can provide them better support in future similar situations. Methods: A total of 380 out of 1305 nurses participated in the survey during December 2020. Coping Inventory for Stressful Situations (CISS) was used. Stepwise regression analysis was used to determine the interaction between sociodemographic characteristics and coping strategies. Results: The CoV married nurses (62.2%) used problem- (*p* = 0.010) and emotion- (*p* = 0.003) focused coping more and avoidance coping less (*p* = 0.007). CoV nurses with master’s degrees (11.1%) used both problem- and emotion-focused coping less (*p* < 0.01), and older nurses used emotional coping more than the younger nurses (*p* = 0.027), whereas younger nurses used more avoidance coping (*p* < 0.01). CoV nurses without children (41%) used avoidance strategies more than nurses who had 2–3 children (*p* < 0.001). Among non-CoV nurses, less use of emotional coping was recorded in nurses with master’s degrees (4%) than in those with a high school diploma (44.2%) (*p* = 0.002). Avoidance coping was also used more by married non-CoV nurses (79.1%) (*p* < 0.001) and those without children (*p* < 0.001). Conclusions: Sociodemographic factors such as working experience, age, level of education and marital status influenced chosen coping strategies during the health crisis.

## 1. Introduction

Since the World Health Organization (WHO) declared the severe acute respiratory syndrome coronavirus 2 (SARS-Cov2) pandemic on 11 March 2020 that caused COVID-19, healthcare workers (HCW) at the forefront have suffered enormous pressure, causing their physical and mental exhaustion [1]. Long work shifts, high risk of infection, lack of specific skills and protective equipment, frustrations, stigmatization and concern about spreading the virus to their families definitely compromised their health with a high prevalence of burnout syndrome among them [2,3,4]. According to studies in the first months of the pandemic, between 71% and 89% of health workers that were in high-risk situations had suffered psychological symptoms [4,5,6].

Among HCW, nurses form the largest group worldwide and the quality of services provided by them decisively affects the efficiency of the health care system and largely determines the satisfaction level of the patients [7]. The same situation is true for Croatia, with approx. 41332 registered nurses, which make up almost 60% of the total number of HCW. Unfortunately, Croatia has not yet reached the European standard for the number of nurses per 100,000 inhabitants, and there is a continuous shortage of them on the market. It is assumed that there are 4000 nurses needed in the Croatian health care system [8]. The crucial role of nurses was also noticed during previous pandemics such as Severe Acute Respiratory Syndrome coronavirus (SARS-CoV), Middle East respiratory syndrome coronavirus (MERS-CoV) and Influenza A (H1N1) [9,10]. 

This is also emphasized in the International Council of Nurses (ICN) report which says that nurses worldwide are currently experiencing a psychological trauma, which may ultimately cause a direct threat to the nursing profession and health care systems [11]. The ICN admits that stress experienced during the COVID-19 pandemic affects more than 50% of American nurses; 49% of Brazilian nurses report anxiety and 25%, depression; 60% of Chinese nurses feel exhausted, and 90% suffer from anxiety; Spanish nurses reported symptoms of anxiety and growing burnout; and 40% of Israeli nurses were also afraid of providing care for COVID-19 patients [12]. 

The COVID-19 pandemic had more than an impact on nurses’ emotions; their coping strategies have also undergone a change, leading them to adopt more efficient coping strategies to prevent the effect on their mental health [12,13]. Coping strategies by definition represent behavioral and cognitive tactics used to manage crises, conditions and demands that are appraised as distressing. Endler & Parker identify three styles of coping: problem-focused (making efforts to solve the problem), emotion-focused (concentrating on oneself and one’s own emotional experiences) and avoidance-oriented (avoiding the problem by engaging in substitute tasks or seeking social contacts) [14].

A study done before this pandemic with Polish nurses showed that they usually chose active coping, planning, self-distraction, seeking emotional support, positive reframing and development [12]. Another study also done before the pandemic among Polish nurses with the use of a multidimensional inventory to measure coping with stress (COPE) showed that active coping and support seeking strategies were dominant in everyday practice [15]. 

On the other hand, a study carried out in China at the beginning of the pandemic showed that nurses displayed quite intensive reactions to the crisis they were experiencing, concentrating on problem-focused strategies rather than on emotions [16].

Coping strategies are usually individualized and influenced by eternal factors such as cultural and workplace context and by individual components such as personal experiences, education levels and resources available to them in a social context [13,17,18,19]. 

The aim of this study was to compare coping strategies applied by nurses working at COVID-19 (CoV) and non-COVID-19 (non-CoV) hospital departments regarding their sociodemographic characteristics during the first wave of the pandemic. Further, it was aimed at investigating the contribution of sociodemographic characteristics to the use of coping strategies among nurses working at CoV and non-CoV departments, respectively.

We hope that the results will help to define which groups of nurses are more exposed to stress and thus save them from burnout in similar crises. 

## 2. Materials and Methods

### 2.1. Ethical Approval

The Ethics Committee of the University Hospital of Split (Ref.: 500-03/20-01/108; approval date 30 October 2020) and the Ethics Committee of the School of Medicine at the University of Split (Reference: 003-08/20-03/0005; approval date 16 November 2020) confirmed that the study was fully in line with the principles of the Helsinki Declaration on Good Clinical Practice (GCP) and approved its implementation.

### 2.2. Participants and Data Collection

A correlation cross-sectional design was used in this study due to data collection in a single time point. It was conducted among nurses (*n* = 1305) employed at the University Hospital of Split, Croatia, in December 2020. Inclusion criteria: nurses employed at University Hospital of Split who worked during the first wave of the COVID-19 pandemic. Exclusion criteria: use of sick leave and maternity leave during the first wave of the pandemic and incomplete forms.

The link for the online survey was sent to the official e-mail addresses of all 1305 nurses. The online form contained clear instructions to the respondent on the purpose of the research, the anonymity of the research participants was emphasized and it was stated that they answered the questions honestly. It was stated that they need approximately 20 min to participate and that only fully completed questionnaires will be accepted. Pressing the “I agree” button was considered consent to participate in the survey. This was followed by questions dealing with demographic and social characteristics (gender, age, education, marital status, financial status), workplace during the COVID-19 pandemic (with patients infected or not infected with SARS-CoV-2 virus) and working experience in the nursing profession (monthly number of working hours, and standardized tools). After filling out the form, participants had to press the “Submit” button to confirm their participation. The data is automatically saved to an Excel spreadsheet. Google Forms does not register incomplete forms. A two-week deadline to complete the survey was set. Two reminder emails were sent, the first after five days and the second after ten days, with an invitation to participate in the research. The data in the Excel spreadsheet was coded by the researchers and re-checked by the PI (PI is the link between the data and the codebook).

Of the 1305 nurses, 380 fully completed the online survey so the response rate was 29.1%. They were divided into two groups according to the answer to the question “Did you work in the COVID-19 department during the COVID-19 pandemic?” The sampling procedure and response rates are shown in Figure 1.

### 2.3. Study Instruments

Coping Inventory for Stressful Situations (CISS)—In this research, the Croatian version of Endler’s and Parkers’ CISS [20] was used for the purpose of measuring coping in stressful situations. CISS consists of 48 tasks divided into three subscales (coping strategies; problem-oriented, emotion-oriented and avoidance-oriented). Problem-oriented coping is defined as solving a stressful situation. Emotion-oriented coping is defined as focusing on reducing feelings of stress and concentrating on one’s own feelings, whereas avoidance refers to behaviors aimed at avoiding coping with stressful situations [21]. Each subscale consists of 16 statements to which respondents respond with a score from 1 (“generally”) to 5 (“always”). The possible range of responses on each scale can vary from 16 to 80, and a higher score indicates more frequent use of certain coping strategies. The internal consistency of Cronbach’s alpha in the Croatian version of the scale starts from above 0.80, 0.82 and 0.75 [20].

In the research of Grgin, Sorić and Kale on the sample of teachers (1994), the original three-factor structure was not completely confirmed [22]. Whereas the first two factors (problem-oriented and emotion-oriented coping) correspond to the original factors, the third extracted factor corresponded to the part of the original factor called Social Diversion, whereas the second part—Distraction—was not confirmed. The Cronbach-alpha coefficients of the obtained three subscales were satisfactorily high (0.85, 0.79 and 0.71). In other research conducted on a student sample [23], four factors were obtained using CFA, i.e. the factor structure of the questionnaire reported by the authors was replicated. Based on this, four subscales were formed to measure styles: problem-oriented coping, emotion-oriented coping, distraction, and social diversion. The internal reliability coefficients of the Cronbach alpha for each scales were 0.80, 0.82, 0.73 and 0.76. Due to the relatively high and significant correlation between scores on the distraction and social diversion subscales, Endler and Parker treat both subscales as one that measures the avoidance-coping strategy. After the formation of such a unique scale, Cronbach’s alpha in the Croatian sample was 0.80. The adapted scales also had satisfactory test-retest reliability coefficients of 0.60 for the problem-oriented subscale, 0.61 for the emotion-oriented subscale and 0.71 for the avoidance-oriented subscale.

### 2.4. The Power of Study

The expected minimum number of subjects for the test power of 0.8 and a 95% confidence interval was a total of 2 × 162 (324) subjects for each observed group (dichotomous endpoint, study of two independent samples). A total of 380 respondents participated in the study; CoV—*n* = 217 subjects and non-CoV—*n* = 163 subjects.

### 2.5. Outcomes of the Study and Hypotheses

The main outcome of the study is to identify the relationship between demographic characteristics and coping strategies during a pandemic between the nurses who worked in the CoV department and nurses who worked in the non-CoV departments.

The hypotheses of the study were as follows:

**Hypothesis 1** **(H1).**
*More efficient coping strategies (problem-focused and emotion-focused) will be used by nurses employed at both CoV and non-CoV departments who are older, not married, have no children, have more working experience and have no professional experience.*


**Hypothesis 2** **(H2).**
*Older age, more professional experience, single, childless and higher education will predict more efficient (problem-focused and emotion-focused) coping in nurses working at CoV and non-CoV departments.*


**Hypothesis 3** **(H3).**
*Younger age, less professional experience, married, have children and less nursing education will contribute to more use of avoidance-coping strategies in nurses working at CoV and non-CoV departments.*


### 2.6. Statistical Analysis

The data were analyzed using Statistical Package for Social Sciences software, version 21 (IBM SPSS Corp, Armonk, NY, USA) for data statistics. Average values of variables were described using the mean (M) and standard deviation (SD). After the descriptive statistical analyses, the t-test or one-way ANOVA with Tukey’s honest significance test (HSD) was used to examine any inter-group differences in coping strategies across the nurses’ sociodemographic variables. The t-test was used to determine whether two groups within a nominal variable (marital status) were statistically different from each other, whereas ANOVA was used to explore whether three or more groups within nominal variables (age, professional experience, number of children and nursing education level) were statistically different from each other. First-order correlations among all variables were explored using the Pearson correlation coefficient. Stepwise regression analysis based on Pearson correlation coefficients was used to determine the interaction between sociodemographic characteristics and coping strategies wherein demographic variables were used as predictor variables, whereas coping strategies were used as a criterion. Using of the stepwise regression enabled testing the addition of each predictor variable using a chosen model fit criterion, adding the variable whose inclusion gives the most statistically significant improvement of the fit and repeating this process until none improves the model to a statistically significant extent. Among the demographic variables, marital status, number of children and educational level were treated as nominal variables, whereas age was treated as a continuous variable. With respect to marital status, participants were divided in two groups comprising those who were married or not married at the time of the research. *p*-values of less than 0.05 were considered statistically significant. There were no missing data in data set.

## 3. Results

The first group included a sample of nurses working at a CoV department at the time of the research (*n* = 217). Most of participants were women (89.9%), aged 33.2 years, who reported an average of 11.6 years of working experience. Most of them were married (62.2%), whereas 30.9% were single or divorced (6.9%) and had no children (41%), and others had one (20.9%) or two (29%) and three (9.2%) children. An equal number of nurses had a high school diploma (44.2%) or a bachelor degree (44.7%), and the least number had a master’s degree (11.1%).

The second group consisted of 163 nurses of both genders (96.3% female), aged 42.1 years, who were working in non-CoV departments at the time when the research took place. On average, the participants in this group had 21 years of working experience. Similar to their counterparts working in CoV departments, most of them were married (79.1%), whereas others were single (12.3%) or divorced (8.6%) and have mostly two (41.7%) children. There was a higher prevalence of nurses with a high school diploma (44.2%) or a bachelor’s degree (52.1%), and less than 4% had a master’s degree (Table 1).

Testing the differences between sociodemographic characteristics and the use of coping strategies among CoV nurses (Table 2) indicated that CoV nurses differed from each other in the use of all coping strategies with regard to marital status and education. The independent sample t-test found that married nurses used more problem- (*p* = 0.010) and emotion- (*p* = 0.003) focused coping and less avoidance coping (*p* = 0.007) in relation to those who were not married. Multiple post-hoc comparisons indicated that the nurses with master’s degrees used both problem- and emotion-focused coping significantly less than the nurses in other educational groups (*p* < 0.001), whereas the nurses with a high school diploma used avoidance coping significantly more than their colleagues with bachelor’s degrees (*p* = 0.031).

Further, nurses in the oldest age category used emotional coping significantly more than the younger nurses (*p* = 0.027), whereas younger nurses, including those with the least working experience, used avoidance coping significantly more than nurses in older age categories (*p* < 0.01). Nurses with 6–15 years of working experience used problem-focused coping more than their colleagues with less working experience (*p* = 0.011). The CoV nurses also differed in the use of avoidance coping with respect to the number of children, indicating that those without children avoided more than nurses who had 2–3 children (*p* < 0.001).

Regarding non-CoV nurses, multiple comparisons found less use of emotional coping among nurses with master’s degrees than among nurses with a high school diploma (*p* = 0.002) and bachelor’s (*p* = 0.012) degrees. Avoidance coping was used more by the married (*p* < 0.001) nurses, those without children compared to those with two or three children (*p* < 0.001) and also by the nurses with bachelor’s degrees compared to the nurses with a high school diploma (*p* < 0.001). Nurses with completed master’s degrees did not differ in the use of avoidance coping, neither from nurses with a high school diploma (*p* = 0.377) nor from those with bachelor’s degrees (*p* = 0.790).

Correlation analyses were conducted to investigate the relationship among the explored variables in CoV and non-CoV nurses (Table 3).

As can be expected, binary correlations, in both groups of nurses, showed that sociodemographic characteristics such as age, marital status, number of children as well as working experience and marital status were mostly interrelated, indicating that older nurses were more often married, had more children and more working experience.

In CoV nurses, age and marital status were mainly positively associated either with problem- or emotion-oriented strategies and negatively with avoidance, showing that older nurses who are married and have fewer children used more efficient (problem- or emotion-oriented) coping strategies, whereas their younger colleagues, who are not married and have fewer children as well as less working experience, used avoidance- coping strategies more during the pandemic. In the other group of non-CoV nurses, less effective avoidance coping was more often used by more educated married nurses with fewer children.

Next, these associations were explored in more detail using stepwise regression analyses. Several separate stepwise regression analyses were run to identify coping strategies preferred by CoV and non-CoV nurses with regards to gender, age, working experience, marital and nursing educational status as well as number of children. Sociodemographic variables were used as independent variables whereas coping strategies were used as criterion.

For CoV nurses (Table 4), the regression model in the third step explains almost 13% of the variance of problem-focused coping, whereas older age (*p* = 0.029) and being married (*p* < 0.001) were related to higher levels of problem-focused coping, and higher levels of education (*p* < 0.001) and having more children (*p* = 0.019) were related to less problem-focused coping. Nurses’ working experience had no significant effect on problem-focused strategies.

Further, the use of emotion-focused coping was determined by age, number of children, marital and educational status, explaining together almost 13% of the variance in the third step, whereas more use of emotion-focused coping is preferred by nurses who are older (*p* = 0.005), married (*p* < 0.001) and have lower levels of nursing education (*p* < 0.001). The significant effects of number of children to the use of emotion-focused coping were not established.

Finally, more use of avoidance coping was associated with single status (*p* = 0.005), fewer children (*p* = 0.002) and less professional experience (*p* = 0.008), indicating that the nurses who are not married, have no children and have less working experience exhibit more avoidance behaviors. The significant effect of age to the use of avoidance coping was not established. This model in the third step explains about 20% of the variance of avoidance coping (Table 4).

For the non-CoV nurses (Table 5), use of problem-focused coping strategies was related to age, indicating that problem-focused strategies have been more widely used by older nurses (*p* < 0.05). The significant effects of marital status and number of children to the use of problem-focused coping were not established. This model in the second step explains 3.6% of the variance. Further, about 8% of the variance of the emotion-focused coping was explained based on age, marital and educational status and number of children, whereas only education reached the level of significance (*p* = 0.001). The significant effects of age and number of children to the use of emotion-focused coping were not established. The marginal effect of marital status (*p* = 0.052) on the use of emotional coping was determined, indicating a tendency of married nurses to cope with emotions.

Finally, avoidance coping strategies have been preferred by nurses who are married (*p* < 0.001) and have no children (*p* < 0.001). This model explains 21% of the variance (Table 5). The significant effect of age was not established.

## 4. Discussion

This study was aimed at assessing the contribution of sociodemographic features to the use of coping strategies in nurses working in CoV and non-CoV departments. A strength of this study is that it was conducted only among nurses, the biggest professional group among HCW and this is the first study like this provided in Croatia. Our study showed that CoV nurses differed from each other in the terms of use of almost all coping strategies with regard to marital status, age, education and working experience. In general, the findings suggest that being younger, single and with lower levels of nursing education can serve as protective factors from nurses’ emotional engagement and active exposure to stressful situations. Married nurses were more likely to use more effective coping strategies such as problem-oriented and emotion-oriented coping, whereas having more children was associated with less use of problem-oriented coping.

We also obtained similar results in a group of nurses employed in non-CoV wards: younger age was associated with less use of problem-oriented coping. Further, married and childless nurses were more likely to use avoidance coping, whereas nurses with higher levels of education were less likely to use emotion-focused coping.

Working during the COVID-19 pandemic was a particular challenge for nurses due to intensified stress and fear of the unknown, especially at the beginning of the pandemic itself [24]. It was already shown that choosing appropriate coping strategies was very important in maintaining good mental health and psychological well-being among healthcare staff. During the SARS epidemic in 2004 and 2005, medical staff in Hong Kong used problem solving strategies rather those focusing on emotions [25].

In line with our results, Sagherian et al. [26] showed that in the groups of nurses working with patients infected with the SARS-CoV-2 virus, the most frequently chosen strategies of coping with stress were strategies focused on the problem as well as emotion-focused strategies. They also showed that nurses working with patients infected with the virus were younger and at the same time had shorter professional experience than nurses working with patients not infected with the SARS-CoV-2 virus, which is similar to our case, where often avoidance coping was used more. On the other hand, our younger non-CoV nurses used more emotional coping probably due to their lack of experience, resources or supervision [12]. Thus, our results addressed that nurses working at non-CoV departments also needed attention and support to minimize the development of posttraumatic stress disorder (PTSD), which is in line with a study by Xiong et al. [27].

A study on Croatian nurses also shows that in the time of the COVID-19 pandemic, nurses use the avoidance and positive reappraisal coping style much more often than physicians do. Whereas physicians first use a strategy of planned and analytical approach to the problem (stressor), nurses first resort to positive reassessment. Furthermore, with respect to age groups, the study shows that individuals under 40 use avoidance coping techniques more often [28]. In our study, the significant effect of age on the use of avoidance coping was not established, although we found more often the use of avoidance coping in nurses with less professional experience.

Another study from Croatia showed that, generally, the most common coping strategies in nurses were problem-focused strategies, then emotion-focused strategies and, most rarely, avoidance coping strategies. The author also showed that higher levels of education was correlated with a higher search for meaning and more common use of active coping, planning and emotional support as coping strategies [29]. This is opposite to our findings, which can be explained with the fact that this study was conducted before the COVID 19 pandemic.

A study from Spain also showed that being older, not being single, living in an independent house and having more than 15 years of working experience protected against stressors and perceived emotions, and were associated with a greater use of coping techniques [30]. They also found a greater impact of perceived negative emotions among auxiliary nurses than among university graduated nurses, which is in line with our findings that CoV nurses with high-school diplomas used avoidance coping significantly more than their colleagues with bachelor’s degrees.

Trumelo et al. also conducted a similar study between HCW in Italy and showed a significant difference in the distribution of perceived stress, anxiety, depression, burnout and secondary trauma levels between HCW who worked with patients affected by the COVID-19 disease [31].

Our study has several limitations. First, it was a cross-sectional study that was conducted in a relatively short period in only one Croatian hospital, which limits the ability to interpret the causal relationships between the different variables in this study. Second, we adopted the strategy of distributing the questionnaire online due to the limitations of social contacts; thus, the study was conducted only in a group of people using information and communication technologies which may have affected the response rate (29.1%). Third, the study sample was only chosen from one city. In order to generalize our results, future longitudinal studies should be conducted using randomized sampling.

## 5. Conclusions

This study clearly showed that sociodemographic factors such as working experience, age, level of education and marital status have an influence on chosen coping strategies during the health crisis. Nurses working in CoV departments choose both active- and emotion-focused coping, whereas those working in non-CoV departments prefer strategies focused on the problem. We hope that our results will encourage the health care system to give special attention to nurses’ working conditions and experience, helping them to choose the best coping strategies to protect their mental health and prevent their burnout.

## Figures and Tables

**Figure 1 healthcare-10-01144-f001:**
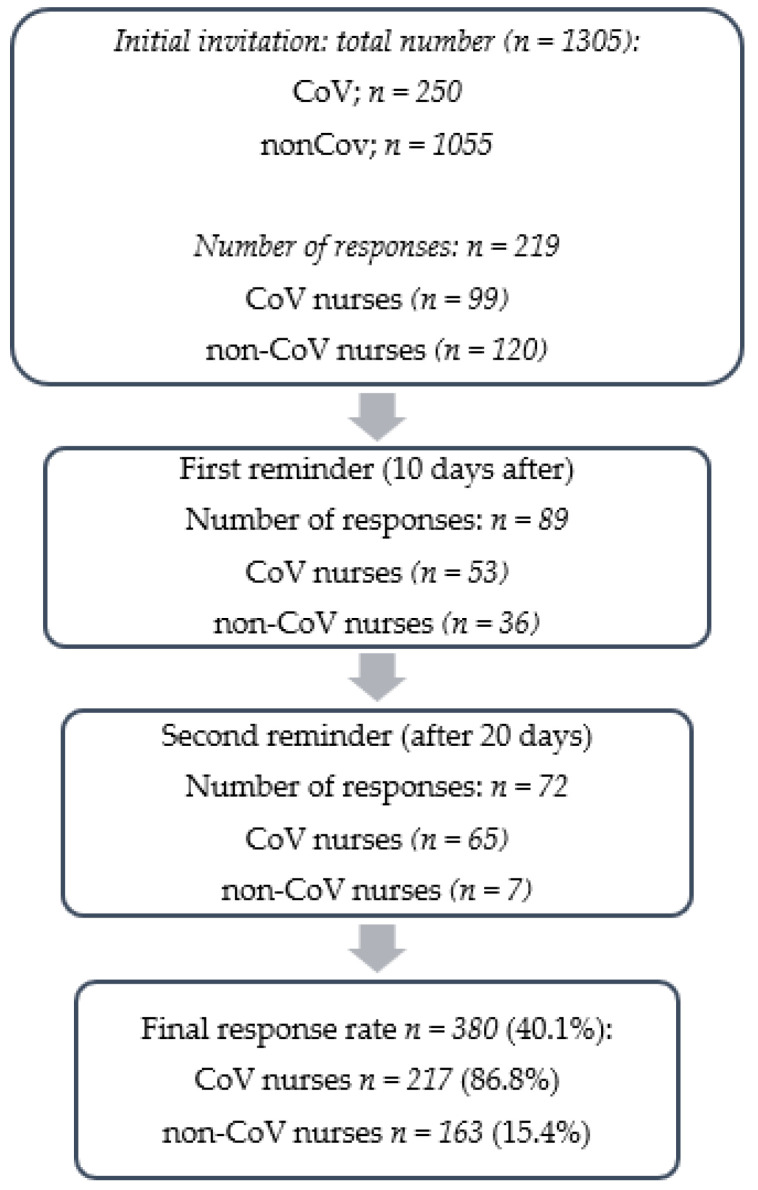
Flow chart of the study.

**Table 1 healthcare-10-01144-t001:** Demographic data of the nurses.

Characteristics		CoV Department (*n* = 217)	Non-CoV Department (*n* = 163)	Total (*n* = 380)
Age (years)		33.15 ± 9.12	42.13 ± 9.53	37.00 ± 10.30
Gender	Female Male	195 (89.9%) 22 (10.1%)	157 (96.3%) 6 (3.7%)	352 (92.6%) 28 (7.4%)
Working experience (years)		11.59 ± 7.80	21.00 ± 9.68	15.63 ± 9.82
Marital status	Not married Married Divorced Widowed	67 (30.9%) 135 (62.2%) 15 (6.9%) 0 (0%)	20 (12.3%) 129 (79.1%) 14 (8.6%) 0 (0%)	87 (22.9%) 264 (69.5) 29 (7.6%) 0 (0%)
Number of children	0 1 2 3	89 (41.0%) 45 (20.7%) 63 (29.0%) 20 (9.2%)	31 (19.0%) 32 (19.6%) 68 (41.7%) 32 (19.6%)	120 (31.6%) 77 (20.3%) 131 (34.5%) 52 (13.7%)
Education degree	High school Bachelor’s degree Master’s degree	96 (44.2%) 97 (44.7%) 24 (11.1%)	72 (44.2%) 85 (52.1%) 6 (3.7%)	168 (44.2%) 182 (47.9%) 30 (7.9%)

Abbreviations: CoV—COVID-19 department; Non-CoV—Non-COVID-19 department.

**Table 2 healthcare-10-01144-t002:** Differences in coping strategies regarding sociodemographic characteristics in CoV and non-CoV nurses.

CoV Department						Non-CoV Department				
Coping Strategies			Mean	SD	F/t	*p*	Mean	SD	F/t	*p*
Problem-focused coping	Age (years)	22–29	3.952	0.544	1.469	0.224	3.631	0.752	2.307	0.079
		30–39	3.812	0.528			3.681	0.803		
		40–49	3.920	0.579			3.969	0.523		
		≥50	4.067	0.161			3.899	0.642		
	Professional experience (years)	0–5	3.879	0.469	3.430 *	0.018	3.400	0.559	2.193	0.091
		6–15	4.026	0.556			3.882	0.656		
		16–25	3.732	0.574			3.961	0.643		
		≥26	3.867	0.184			3.783	0.639		
	Marital status	Yes	4.012	0.559	2.593 **	0.010	3.905	0.630	0.933	0.352
		No	3.827	0.491			3.810	0.668		
	Number of children	0	4.016	0.556	2.062	0.106	3.755	0.636	0.544	0.653
		1	3.855	0.460			3.858	0.525		
		2	3.828	0.521			3.927	0.706		
		3	3.823	0.544			3.821	0.657		
	Nursing educational level	High school	3.922	0.487	11.989 ***	0.000	3.791	0.692	0.858	0.426
		Bachelor degree	4.012	0.559			3.905	0.629		
		Master Degree	3.450	0.283			4.044	0.0344		
Emotion-focused coping	Age (years)	22–29	2.668	0.602	2.762 *	0.043	2.794	0.446	1.057	0.369
		30–39	2.778	0.929			3.086	0.638		
		40–49	2.832	0.717			2.814	0.751		
		≥50	3.223	0.210			2.982	1.105		
	Professional experience	0–5	2.727	0.539	0.762	0.517	2.844	0.273	0.254	0.859
		6–15	2.810	0.822			2.955	0.663		
		16–25	2.704	0.853			2.948	0.683		
		≥26	3.014	0.404			2.833	1.0856		
	Marital status	Yes	2.939	0.706	3.007 **	0.003	2.86	0.776	−0.785	0.433
		No	2.642	0.735			2.96	0.856		
	Number of children	0	2.694	0.707	1.040	0.376	2.724	0.551	1.241	0.297
		1	2.861	0.679			3.0293	0.749		
		2	2.864	0.741			2.991	0.91		
		3	2.659	0.945			2.787	0.86		
	Nursing educational level	High school	2.747	0.671	9.971 ***	0.000	3.049	0.826	6.241	0.002
		Bachelor degree	2.939	0.706			2.86	0.776		
		Master Degree	2.224	0.841			1.896	0.355		
Avoidance coping	Age	22–29	3.848	0.5123	14.471 ***	0.000	3.257	0.412	0.092	0.965
		30–39	3.311	0.534			3.195	0.65		
		40–49	3.241	0.856			3.247	0.598		
		≥50	3.496	0.621			3.267	0.641		
	Professional experience	0–5	3.836	0.444	9.546 ***	0.000	3.344	0.174	1.021	0.385
		6–15	3.507	0.643			3.243	0.584		
		16–25	3.204	0.786			3.316	0.605		
		≥26	3.423	0.612			3.131	0.636		
	Marital status	Yes	3.402	0.756	−2.734 **	0.007	3.456	0.571	5.098	0.000
		No	3.645	0.551			3.01	0.542		
	Number of children	0	3.773	0.54	10.698 ***	0.000	3.617	0.584	7.995	0.000
		1	3.586	0.703			3.373	0.483		
		2	3.306	0.62			3.119	0.567		
		3	3.097	0.742			3.014	0.607		
	Nursing educational level	High school	3.641	0.559	3.729 *	0.026	2.986	0.532	13.953	0.000
		Bachelor degree	3.402	0.756			3.456	0.571		
		Master Degree	3.662	0.528			3.302	0.622		

SD-standard deviation, F/t *-ANOVA/*t*-test, * *p* < 0.05, ** *p* < 0.01, *** *p* < 0.001.

**Table 3 healthcare-10-01144-t003:** Correlation matrix for the tested variables.

	Age	Marital Status	Number of Children	Working Experience	Eductional Level	Problem-Oriented Coping	Emotion-Oriented Coping	Avoidance Coping
Age	-	0.212**	0.616 **	0.806 **	0.390 **	0.055	0.183 **	−0.235 **
Marital status	−0.086	-	−0.041	0.150 *	0.570 **	0.215 **	0.236 **	−0.168 *
Number of children	0.248 **	−0.307 **	-	0.697 **	0.170 *	−0.148 *	0.028	−0.347 **
Working experience	0.954 **	−0.098	0.300 **	−	0.385 **	−0.072	0.085	−0.338 **
Educational level	−0.057	0.833 **	−0.367 **	−0.074	-	−0.109	−0.096	−0.090
Problem-focused coping	0.115	0.083	0.037	0.063	0.111	-	0.194 **	0.451 **
Emotion-focused coping	−0.060	−0.123	0.047	−0.011	−0.255 **	−0.012	-	0.200 **
Avoidance coping	−0.035	0.403 **	−0.358 **	−0.074	0.408 **	0.419 **	0.095	-

Above diagonal—CoV department, below diagonal—non CoV departments, * *p* < 0.05. ** *p* < 0.01.

**Table 4 healthcare-10-01144-t004:** Results of the stepwise regression analysis using coping strategies as a criterion in CoV nurses.

Criterion	Predictors	Step 1			Step 2			Step 3		
		*β*	*t*	*p*	*β*	*t*	*p*	*β*	*t*	*p*
Problem- focused coping	Age (years)	0.035	0.508	0.612				0.178	2.204	0.029
	Marital status (Y/N)				0.116	1.410	0.160	0.264	3.584	0.000
	Number of children				0.140	2.015	0.045	−0.184	−2.367	0.019
	Educational level				−0.210	−2.613	0.010	−0.305	−4.079	0.000
		R = 0.035 R^2^ = 0.001 F (1.215) = 0.258 *p* = 0.612			R = 0.246 R^2^ = 0.060 F (3.213) = 4.569 * *p* = 0.004			R = 0.359 R^2^ = 0.129 F (4.212) = 7.839 * *p* = 0.000		
Emotion- focused coping	Age (years)	0.185	2.753	0.006	0.166	2.014	0.045	0.228	2.813	0.005
	Marital status				0.162	2.338	0.020	0.285	3.879	0.000
	Number of children				−0.033	−0.412	0.681	−0.007	−0.096	0.924
	Educational level							−0.304	−4.057	0.000
		R = 0.185 R^2^ = 0.034 F (1.215) = 7.577 * *p* = 0.006			R = 0.248 R^2^ = 0.062 F (3.213) =4.657 * *p* = 0.004			R = 0.359 R^2^ = 0.129 F (4.212) =7.860 * *p* = 0.000		
Avoidance coping	Age (years)	−0.255	−3.862	0.000	−0.019	−0.239	0.811	0.168	1.621	0.106
	Marital status				−0.193	−2.951	0.004	−0.184	−2.855	0.005
	Number of children				−0.358	−4.740	0.000	−0.264	−3.199	0.002
	Working experience							−0.300	−2.665	0.008
		R = 0.255 R^2^ = 0.065 F (1.215) = 14.917 * *p* = 0.006			R = 0.411 R^2^ = 0.169 F (3.213) = 14.464 * *p* = 0.000			R = 0.443 R^2^ = 0.196 F (4.212) = 12.934 * *p* = 0.000		

* *p* < 0.05.

**Table 5 healthcare-10-01144-t005:** Results of the stepwise regression analysis using coping strategies as a criterion in non-CoV nurses.

Criterion	Predictors	Step 1			Step 2			Step 3		
		*β*	*t*	*p*	*β*	*t*	*p*	*β*	*t*	*p*
Problem-focused coping	Age (categories)	0.164	2.113	0.036	0.163	2.008	0.046			
	Marital status				0.099	1.202	0.231			
	Number of children				0.034	0.393	0.695			
		R = 0.164 R^2^ = 0.027 F (1.161) = 4.465 * *p* = 0.036			R = 0.189 R^2^ = 0.036 F (3.159) =1.965 * *p* = 0.121					
Emotion-focused coping	Age	−0.075	−0.961	0.338	−0.091	−1.106	0.271	−0.064	−0.800	0.425
	Marital status				−0.059	−0.709	0.479	0.228	1.959	0.052
	Number of children				0.035	0.410	0.682	−0.038	−0.445	0.657
	Educational level							−0.409	−3.404	0.001
		R = 0.075 R^2^ = 0.006 F (1.161) = 0.923 *p* = 0.338			R = 0.107 R^2^ = 0.011 F (3.159) = 0.616 *p* = 0.606			R = 0.281 R^2^ = 0.079 F (4.158) = 3.390 * *p* = 0.011		
Avoidance coping	Age	−0.001	−0.013	0.989	0.111	1.515	0.132			
	Marital status				0.290	3.912	0.000			
	Children				−0.297	−3.845	0.000			
		R = 0.001 R^2^ = 0.000 F (1.161) =0.000 *p* = 0.989			R = 0.462 R^2^ = 0.213 F (3.159) =14.364 * *p* = 0.000					

* *p* < 0.05.

## Data Availability

The data presented in this study are available upon request of the respective authors. Due to the protection of personal data, the data are not publicly available.

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
