# Peer review of "The Impact of Sociodemographic Characteristics on Coping Strategies Used by Nurses Working at COVID and Non-COVID Hospital Departments during COVID-19 Pandemic: A Cross-Sectional Study"

_healthcare, 2022, doi:10.3390/healthcare10061144_

Round 1
Reviewer 1 Report
This study conducts a survey to collect nurses’ coping strategies at Covid and non-Covid hospital departments and examine the associations with their sociodemographic characteristics in Croatia. The idea is interesting and the paper is well-organized. I have some comments as follows:
Major comments:
In the abstract section:
The background only tells the purpose of doing this study and needs to add one more sentence to say the background although the covid pandemic has been well known to people.
The methods need to be specific, for example, add 380 of 1305 participated in the survey of what coping strategies, and what kinds of statistical analysis are used on what variables. Otherwise, the description of the results will be confusing.
The results should be: “Among CoV nurses, married nurses used more problem-focused and emotion-focused, and less avoidant coping strategies, all are statistically significant at 0.05”, or something better to clarify the findings.
Introduction:
Add some background/introductions of the roles of nurses in Croatia in taking care of the patients, it would be helpful and supportive of the theme of this paper as readers like me may think that primary care physicians rather than nurses are the first one patients would contact once they need some cares.
Materials and Methods:
The 2.5. Statistical analysis section needs some improvements, such as how to differentiate older vs. younger, which reference group is chosen for t-test or one-way ANOVA, how stepwise regression is run after selecting the variables based on the Pearson correlation coefficient.
Results:
Table 1 is described but not mentioned in the lines 147-160, at least need to say: Table 1 reports …
Line 171, it should be “p<0.001” considering the p values are zero in Table 2, right?
Lines 171-172, I didn’t see that in Table 2 if I understand it correctly.
Lines 174-178, I didn’t see the three p values in Table 2, do I miss anything here?
Lines 184-185, the sentences say something opposite based on the p-values in Table 2. For example, avoidant coping is used more by non-married rather than married given the positive F/t values with p=0.000.
In Table 3, what do the “1, 2, 3, …, 8” mean? If referring to Age, Marital status, …., and avoidant coping, be specific please. Also, Table 3 needs some descriptions rather than put it there without any illustrations.
Also, in Table 3, the marital status is highly correlated with education level in non-CoV department but in Table 5 for emotion-focused coping, both variables are retained, how could this occur? The selection criterion of variables should be specific before running stepwise regressions. Double check thoroughly about the variables used in Tables 4-5 based on Table 3.
Lines 232-233, the sentence causes confusion as it reads like only the nurses with less education explain the variability, instead, it should be the combination of four variables explain 8% of the variance of emotion-focused coping strategy.
In Tables 4 and 5, how could the authors differentiate the older vs. younger, married vs. single, etc.? I only read the nominal variables in section 2.5.
Discussion:
Lines 251-252, the sentence is incorrect as education does not contribute to the avoidant coping according to Table 4.
Lines 252-255, Table 4 does not tell working experience contributes to the emotion-focused coping.
Line 274, the interpretation from Table 4 is incorrect as line 213 says older nurses are more likely to use emotion-focused coping but Table 5 shows younger nurses are more likely to use emotion-focused coping in Non-CoV department.
Lines 283, neither the Table 4 nor 5 gives this finding.
Lines 285-289, seems like the findings from that study contradict with what the authors found in this study. If yes, add some explanations.
Lines 294-296, if I understand it correctly, this comes from Table 2, if yes, add the reference group as the table does not explicitly tell which one the reference.
Lines 306-307, the limitation should be extended to one hospital as only one hospital was surveyed.
Minor comments:
Do grammar check throughout the paper, some changes are detected:
Line 51, it is negative? I think it is positive.
Line 66, contex -> context?
Line 214, p should be 0.005 according to Table 4.
Line 221, it should be the third step rather than the second.
Do grammar check for lines 251-257, 266.
Author Response
Dear reviewer, we thank you for valuable comments and suggestions that helped us improve our manuscript

Reviewer 2 Report
-
The claim needs to be formulated more clearly!
More explanations are required about the type of studio chosen (cross-sectional).
Factors must be described in methods.
The concepts of avoiding strategy should be explained.
The implications of the results must be clearly stated. What will be done with the results and who they will use?
The conclusions must be drawn from the point of view of the claim.
There are some typos. Rows 252-257 should be rewritten to make them easier to understand.
-
Author Response

(The authors gave the same response as above.)

Reviewer 3 Report
The authors investigated coping strategies applied by nurses working at COVID (CoV) and non COVID (non-CoV) hospital departments regarding their sociodemographic characteristics during the COVID-19 pandemic.
Paper is interesting, well designed with appropriate methodology and statistical analysis. As it deals with an important topic which has been reported only in a few very limited studies it is worthy of publication. However, I have several objections that should be revised prior to possible publication of the paper:
1. Abstract should be updated – In the methodology section of the abstract the authors should clearly state outcomes of the study, as well as to present the main methodology used for the purpose of this research (which instruments – tests were used...).
2. Also in abstract the author’s should present calculated values (n or percentage) for each investigated variable, not only p-values as they did.
3. Line 37. – 'symptom' should be 'symptoms'
4. Please provide the full title of each abbreviation used in text. E.g. some abbreviations such as ''SARS-CoV'', ''MERS-CoV'', ''COVID-19'' or ''HCW'' have not been explained at the place where they were first mentioned in text. Please revise.
5. As the coping strategies are the main outcome of this article the authors should add a few lines in introduction explaining the definition of coping strategies, list some common coping strategies and explain the importance of coping strategies among health workers.
6. Line 81/82 ... It was conducted among nurses (1305)... Please add ''n = 1305'' otherwise some may think that the number in brackets represents reference.
7. Please add a new paragraph in methodology ''Outcomes of the study and hypothesis'' and explain primary and secondary outcomes of the study, as well as hypothesis.
8. The authors used Croatian versions of several scales. Please add info whether these scales have been validated?
9. Which statistical test was used to test normality of distribution of the data? Please update!
10. Table 3 – What the numbers 1-8 in the first line of the Table represents. It should be clear from the Table. Please revise.
Author Response
Dear reviewer, we thank you for valuable comments and suggestions that helped us improve our manuscript

This manuscript is a resubmission of an earlier submission. The following is a list of the peer review reports and author responses from that submission.